# Evaluation of the Antifungal, Antioxidant, and Anti-Diabetic Potential of the Essential Oil of *Curcuma longa* Leaves from the North-Western Himalayas by *In Vitro* and *In Silico* Analysis

**DOI:** 10.3390/molecules27227664

**Published:** 2022-11-08

**Authors:** Nitin Sharma, Nidhi Gupta, Raha Orfali, Vikas Kumar, Chirag N. Patel, Jiangnan Peng, Shagufta Perveen

**Affiliations:** 1Department of Biotechnology, Chandigarh College of Technology, CGC, Landran, Mohali 140307, India; 2Department of Pharmacognosy, College of Pharmacy, King Saud University, Riyadh 11495, Saudi Arabia; 3University Institute of Biotechnology, Chandigarh University, Gharuan, Mohali 140413, India; 4Department of Botany, Bioinformatics, and Climatic Change Impacts Management, School of Sciences, Gujarat University, Ahmedabad 380009, India; 5Department of Medicinal, School of Computer, Mathematical and Natural Sciences, Morgan State University, Baltimore, MD 21251, USA

**Keywords:** *Curcuma longa*, essential oil, GC–MS, antifungal, antioxidant, anti-diabetic, molecular docking, MD simulations, MM-GBSA, toxicity

## Abstract

Essential oils (EOs) have gained immense popularity due to considerable interest in the health, food, and pharmaceutical industries. The present study aimed to evaluate the antimicrobial and antioxidant activity and the anti-diabetic potential of *Curcuma longa* leaf (CLO) essential oil. Further, major phytocompounds of CLO were analyzed for their *in-silico* interactions with antifungal, antioxidant, and anti-diabetic proteins. CLO was found to have a strong antifungal activity against the tested Candida species with zone of inhibition (ZOI)-11.5 ± 0.71 mm to 13 ± 1.41 mm and minimum inhibitory concentration (MIC) was 0.63%. CLO also showed antioxidant activity, with IC_50_ values of 5.85 ± 1.61 µg/mL using 2,2-diphenyl-1-picrylhydrazyl (DPPH) scavenging assay and 32.92 ± 0.64 µM using ferric reducing antioxidant power (FRAP) assay. CLO also showed anti-diabetic activity with an IC_50_ of 43.06 ± 1.24 µg/mL as compared to metformin (half maximal inhibitory concentration, IC_50_-16.503 ± 0.66 µg/mL). Gas chromatography–mass spectrometry (GC–MS) analysis of CLO showed the presence of (-)-zingiberene (17.84%); 3,7-cyclodecadien-1-one, 3,7-dimethyl-10-(1-methylethylidene)-(15.31%); cyclohexene, 4-methyl-3-(1-methylethylidene) (12.47%); and (+)-4-Carene (11.89%) as major phytocompounds. Molecular docking of these compounds with antifungal proteins (cytochrome P450 14 alpha-sterol demethylase, PDB ID: 1EA1, and N-myristoyl transferase, PDB ID: 1IYL), antioxidant (human peroxiredoxin 5, PDB ID: 1HD2), and anti-diabetic proteins (human pancreatic alpha-amylase, PDB ID: 1HNY) showed strong binding of 3,7-cyclodecadien-1-one with all the selected protein targets. Furthermore, molecular dynamics (MD) simulations for a 100 ns time scale revealed that most of the key contacts of target proteins were retained throughout the simulation trajectories. Binding free energy calculations using molecular mechanics generalized born surface area (MM/GBSA), and drug-likeness and toxicity analysis also proved the potential for 3,7-cyclodecadien-1-one, 3,7-dimethyl-10-(1-methylethylidene) to replace toxic synthetic drugs and act as natural antioxidants.

## 1. Introduction

Natural products have become an alternative and supplementary treatment technique due to their diverse pharmacological and biological applications [1,2]. Medicinal plants include a variety of phytoconstituents and have a wide range of pharmacological characteristics. Plants have long been utilized as natural remedies for asthma, colds, fevers, coughs, cholera, and diarrhea [3]. Approximately 80% of the population in developing nations relies on traditional plant-based remedies [4,5]. Traditional medical practices were found to have greater efficacy, fewer adverse effects, lower costs, and greater accessibility than modern medical practices [6]. In addition to pharmaceutical therapies and herbal beverages, natural compounds such as essential oils have been used in traditional and modern medicines, perfumes, and cosmetics [7,8]. Several EOs and phytocompounds are considered “Generally Recognized as Safe” (GRAS) [9,10]. EOs have been studied for their antimicrobial and antioxidant activities, which have been found to be the most common biological activities. However, some other activities of EOs, such as their antiviral, insecticidal, angiotensin-converting enzyme, amylase, and glucosidase-enzyme-inhibiting effects, still require more research. More research needs to be conducted in order to figure out how antimicrobial and antioxidant activity works and what phytocompounds are responsible for these effects.

*Curcuma* L. (Zingiberaceae) is a family of perennial rhizomatous herbs indigenous to tropical and subtropical regions. This genus is widely cultivated in tropical and subtropical regions of Asia, Australia, and South America [11]. There are approximately 93–100 accepted *Curcuma* species; however, the exact number of species is still controversial [12]. The genus is renowned for its importance as a source of coloring and flavoring agents in Asian cuisines, traditional medicines, spices, dyes, perfumes, cosmetics, and ornamental plants [13]. Several *Curcuma* species are used medicinally to treat pneumonia, bronchial complaints, leucorrhea, diarrhea, dysentery, infectious wounds or abscesses, and insect bites [12,14] in Bangladesh, Malaysia, India, Nepal, and Thailand [14]. Turmeric (*Curcuma longa*), a member of the *Zingiberaceae* family, has attracted much attention for producing a large number of complex compounds that are beneficial in foods such as spices, flavoring, and seasoning, as well as in the cosmetic and pharmaceutical industries [15,16]. Usually, utilization of *C. longa* is limited to rhizome, and its therapeutic properties include insecticidal [17,18,19], antimicrobial [20,21,22], antimalarial [23,24], antiviral [25,26,27], and antioxidant properties [18,28,29]. The dried rhizome of *C. longa* is also used as a food color in curry powder in Asian countries [30]. Its leaves have also been used as a spice in Malaysia and India [31]. Leaves of *C. longa* are reported to have good quantity of proteins, carbohydrates, fibers, and ash [31]. According to studies, these leaves are a good source of bioactive compounds that can protect individuals from several diseases such as cancer and premature ageing [32,33]. In spite of their high nutritional potential, leaves of *C. longa* are discarded after rhizome cultivation in Brazil and India.

Modern drug design uses molecular docking tools to understand the drug–receptor interaction [34]. By elucidating the drug–receptor interaction mechanism, computational approaches encourage and facilitate the advent of new, more potent inhibitors. The Swiss ADME and PROTOX web tools, as a free software program, can be used to predict the physicochemical characteristics, absorption, distribution, metabolism, elimination, and pharmacokinetic features of molecules, being crucial endeavors for subsequent clinical trials [35,36]. Therefore, ligand–protein docking can predict the predominant binding model of a ligand with a target protein of known three-dimensional structure [37]. The main objective of the present study was to identify potential phytocompounds (ligands) of the essential oil from *C. longa* leaves for its antifungal, antioxidant, and anti-diabetic potential using in vitro and in silico techniques. This could be investigated further as a potential therapeutic intervention for antifungal and anti-diabetic drugs, with the goal of generating a novel drug that has a high chance of success while shortening the time required for drug discovery. The novelty of the present study is the utilization of the waste leaves of *C. longa* as source of antifungal and anti-diabetic agents and natural antioxidants. However, further in vivo studies are required to validate these medicinal properties of *C. longa* leaves.

## 2. Results and Discussions

### 2.1. Percentage Yield of CLO

The essential oil from fresh leaves of *C. longa* of the *Zingiberaceae* family from the lower regions of Himachal Pradesh (≈650 m above the sea level) was extracted using the hydro distillation method. The percentage yield of essential oil obtained from leaves was 0.10% (v/w). In contrast to our report, 0.65% of oil was extracted from fresh leaves of *C. longa* obtained from Uttaranchal, India, in September 2000 [38]. A study from Leela et al. [39] reported a 1.3% yield of EO from leaves of *C. longa* from Calicut. Similar to this study, a high yield of 1.45% (v/w) was also reported by Parveen et al. [40]. Essential oil content of leaf samples collected from different regions of Orissa varied from 0.37 to 0.8% [41]. The variations in geographical locations, genotypes, and season of collection are the major factors responsible for variation in extraction yield [42,43,44]. 

### 2.2. Chemical Composition of Essential Oil of Curcuma longa Leaves through GC–MS

GC–MS data were obtained from CIL/SAIF Panjab University, Chandigarh, using helium as a carrier gas. GC–MS analysis of CLO showed the presence of (-)-zingiberene (17.84%); 3,7-cyclodecadien-1-one, 3,7-dimethyl-10-(1-methylethylidene)-(15.31%); cyclohexene, 4-methyl-3-(1-methylethylidene) (12.47%); and (+)-4-carene (11.89%) as major phytocompounds (Table 1, Figure 1). In contrast to our study, Parveen et al. [40] identified eucalyptol (10.27%) as major component of the leaf oil of *C. longa*. Several other compounds such as α-pinene (1.50%), β-phellandrene (2.49%), β-pinene (3.57%), limonene (2.73%), 1,3,8-p-menthatriene (1.76%), ascaridole epoxide (1.452%), 2-methylisoborneol (2.92%), and 5-isopropyl-6-methyl-hepta-3, dien-2-ol (2.07%) were also present in considerable quantity in leaf oil. Chaaban et al. [15] have showed the presence of α-phellandrene (41.99%), ρ-mentha-2,4(8)-diene (24.89%), 1,8-cineole (7.82%), ocimene (2.79%), myrcene (2.63%), and α-pinene (2.52%) in the essential oil of *C. longa*.

### 2.3. Antifungal Activity of CLO against Fungal Strains

Antimicrobial activity was determined in terms of diameter of ZOI (mm) and MIC value (µg/mL). CLO was found to be effective against both tested fungal strains. The diameter of ZOI of CLO was found to be 13.0 ± 1.41 mm and 11.5 ± 0.71 mm against *C. albicans* (MTCC90028) and *C. albicans* (ATCC277), respectively. Fluconazole (25µg) showed a ZOI of 18 ± 0.7 mm and 13 ± 0.71 mm against *C. albicans* (MTCC90028) and *C. albicans* (ATCC277), respectively (Appendix A, Table 2). The MIC of CLO was found to be 0.63% against *C. albicans* (ATCC277) and *C. albicans* (MTCC90028). The MIC of fluconazole was found to be 0.063% for *C. albicans* (MTCC90028) and *C. albicans* (ATCC277), as shown in Table 2. The strong antimicrobial activity of the leaf oil of *C. longa* was also reported against *B. cereus* (MIC-78 μg/mL), *S. aureus* (MIC-78 μg/mL), and *A. niger* (MIC-19.5 μg/mL) by Essien et al. [45]. In another study, Parveen et al. [40] reported the maximum inhibition of leaves oil of *C. longa* against *F. miniformes* MAY 3629 (22 mm), followed by *B. subtilis* ATCC 6633 (21 mm) and *A. flavus* ATCC204304 (20 mm) after 48 h of incubation. The antimicrobial activity of *C. longa* extract has been attributed to compounds belonging to flavonoids and terpenes, particularly to borneol, cymene, cuparene, and careen [46].

### 2.4. In Vitro Antioxidant and Anti-Diabetic Activity of CLO

The antioxidant potential of CLO from leaves of *C. longa* was determined using % DPPH radical scavenging and the ferric reduction capacity (FRAP) method. The antioxidant capacity of CLO was found to be dose dependent (Figure 2). The IC_50_ values of CLO were found to be 5.85 ± 1.61 µg/mL and 32.92 ± 0.64 µM for DPPH and FRAP, respectively, whereas ascorbic acid showed IC_50_ values of 3.11 ± 0.47 µg/mL and 24.09 ± 2.16 µM for DPPH and FRAP, respectively (Table 3). The antioxidant activity of rhizome was reported by several studies [47,48]. However, only a few studies have reported the antioxidant activity in *C. longa* leaves [32,44]. Chan et al. [32] evaluated the antioxidant activity of leaves of *C. longa* in fresh and freeze-dried samples, finding that fresh samples had high ascorbic acid equivalent antioxidant capacity (AEAC) (243 ± 28 mg AA/100 g) and ferric-reducing power (FRP) (2.1 ± 0.1 mg GAE/g), as compared to that of freeze-dried samples (AEAC-222 ± 12 mg AA/100 g; FRP-1.8 ± 0.1 mg GAE/g). A study by Mishra et al. [44] compared the genetic diversity of *C. longa* in different accessions and also compared their biological activities. They reported antioxidant activities in different accessions using DPPH (46.56–87.10%), FRAP (22.13 ± 1.62–204.43 ± 45.84 mmol Fe II/g), nitric oxide scavenging assay (9.18–63.95%), and total antioxidant assays (93.22 ± 5.42–299.92 ± 85.57 mg AAE/g).

The anti-diabetic potential of CLO was evaluated using the α-amylase inhibition method and was found to have an IC_50_ value of 43.06 ± 2.51 µg/mL as compared to that of the standard drug, metformin (16.51 ± 2.11 µg/mL) (Table 3). However, both CLO and metformin were found to show dose-dependent α-amylase inhibition activity (Figure 2). The anti-diabetic activity of *C. longa* rhizome extract or oil was reported in several reports [49,50,51,52]. Fresh (IC_50_-64.7 ± 5.9 µg/mL) and dry rhizomes (IC_50_-34.3 ± 6.2 µg/mL) of *C. longa* were found to have strong α-amylase inhibition as compared to that of acarbose (296.3 ± 12.7 µg/mL) [50]. Kalaycıoğlu et al. [52] evaluated the α-amylase inhibitory activity in three curcuminoids, namely, bisdemethoxycurcumin, demethoxycurcumin, and curcumin, isolated from *C. longa* rhizome with IC_50_ values of 12.5 ± 0.2 µM, 21.1 ± 0.3 µM, and 12.5 ± 0.2 µM, respectively, as compared to that of the standard genistein (2.50 ± 0.02 µM). However, our study is the first report in which the antidiabetic potential of essential oil from leaves of *C. longa* has been shown.

### 2.5. Molecular Docking of Selected Phytocompounds of CLO with Target Antifungal, Anti-Oxidant, and Anti-Diabetic Proteins and MM-GBSA Analysis of Best Docked Ligand

The molecular docking study was conducted in order to study the molecular mechanism of action of major phytocompounds of CLO with fungal proteins (IEA1 and 1IYL), antioxidant protein (1HD2), and diabetic protein (1HNY). Synthetic drugs such as fluconazole, ascorbic acid, and metformin were used as standard control. Among all selected phytocompounds, 3,7-cyclodecadien-1-one was found to show strong binding energy of −21.331 kcal mol^−1^, −24.223 kcal mol^−1^, −19.399 kcal mol^−1^, and −20.819 kcal mol^−1^ against 1EA1, 1IYL, IHD2, and 1HNY proteins, respectively (Table 4). Fluconazole showed binding energy of −37.349 kcal mol^−1^ and −38.248 kcal mol^−1^ with 1EA1 and 1IYL proteins, respectively. Ascorbic acid showed binding energy of −23.999 kcal mol^−1^ against the 1HD2 protein, and metformin showed binding energy of −17.117 kcal mol^−1^ against the 1HNY protein. The binding energy, hydrogen bonds, and interactive amino acids of selected phytocompounds with protein targets and standard drugs are summarized in Table 5. Further interactions of 3,7-cyclodecadien-1-one with 1EA1, 1IYL, IHD2, and 1HNY proteins analyzed using Chimera 1.14 and Discovery Studio Visualizer are shown in Figure 3A–L. Binding interactions of casuarinin were analyzed using Discovery Studio (DS) Visualizer and were found to have six hydrogen bonds with Tyr(A):35, Arg(A):167, Phe(A):182, Tyr(A):184, Asp(A):263, and Gln(A):267 residues of the 4YAY protein (Figure 3A); three hydrogen bonds with Lys(A):249, Ser(A):251, and Arg(A):256 residues of the 4DLI protein (Figure 3B); three hydrogen bonds with Asn(A):567, Arg(A):571, and Glu(A):719 residues of the 1HW9 protein (Figure 3C); and two hydrogen bonds with Ala(A):92 and Asp(A):112 residues of the 1B09 protein (Figure 3D). The non-covalent interactions of casuarinin with target proteins are shown in Figure 3A–D.

### 2.6. MD Simulations

On the basis of molecular docking results, the best ligand–protein complexes were selected for MD simulations. Since, 3,7-cyclodecadien-1-one, 3,7-dimethyl-10-(1-methylethylidene) reported in CLO was found to have the best binding energy with all selected target proteins, complexes of 3,7-cyclodecadien-1-one with 1EA1, 1IYL, IHD2, and IHNY proteins were selected for MD simulations for 100 ns.

#### 2.6.1. Root Mean Square Deviation (RMSD) of Protein–Ligand Complexes

On performing MD simulations, the root mean square deviation (RMSD) is used to measure the average change in displacement of a selection of atoms for a particular frame with respect to a reference frame. It is calculated for all frames in the trajectory. The plots in Figure 4 showed the RMSD evolution of a protein (left Y-axis). The docked pose of ligand and protein as a whole complex is considered as the reference starting frame, and then the movement from this reference position during the MD simulation is measured by aligning all the protein frames obtained during the MD trajectories. Checking the RMSD of the protein can provide knowledge in terms of its auxiliary 3-D structural movement on a graph during the simulation. RMSD examination can demonstrate if the simulation has equilibrated—its changes towards the finish of the recreation are around some thermal energetically stable conformation. Changes in the range of 1–3 Å are completely satisfactory for small globular proteins. However, this range of value widens as the size of the protein increases. The RMSD graph of 3,7-cyclodecadien-1-one, 3,7-dimethyl-10-(1-methylethylidene) in a complex with the 1EA1 protein was found to be stabilized between 1.6 and 3.2 Å from 0 to 100 ns (Figure 4A), while the RMSD of 3,7-cyclodecadien-1-one, 3,7-dimethyl-10-(1-methylethylidene) in complex with the 1IYL protein was found to be stable between 2.5 and 4.0 Å from 0 to 100 ns (Figure 4B). The RMSD of the 3,7-cyclodecadien-1-one, 3,7-dimethyl-10-(1-methylethylidene)–1HD2 protein complex was found to be unstable at 0–65 ns, but became stable between 65 and 85 ns between 1.5 and 2.5 Å (Figure 4C). The RMSD of the 3,7-cyclodecadien-1-one, 3,7-dimethyl-10-(1-methylethylidene)–1HNY complex was found to be stabilized between 2 and 2.5 Å from 0 to 100 ns (Figure 4D). RMSD data revealed the stability of 3,7-cyclodecadien-1-one, 3,7-dimethyl-10-(1-methylethylidene) in the binding pocket of all the selected target proteins.

The ligand RMSD (right Y-axis, plots of Figure 4) suggests the stability of ligand posture concerning the docked position of the ligand in the binding cleft of the protein. For this, the values slightly larger than the protein’s RMSD are considered satisfactory, but if the values observed are significantly larger than the RMSD of the protein, then it is likely that the ligand acquires a different stable position than the original posture. For the 3,7-cyclodecadien-1-one, 3,7-dimethyl-10-(1-methylethylidene)–1EA1 complex (Figure 4A), the Lig fit Prot stayed significantly lower than the protein’s RMSD from 0 to 10 ns and then from 70 to 90 ns during simulation, suggesting slight changes in pose between 10 and 70 ns; thereafter, the orientation of the ligand remained stable. For the 3,7-cyclodecadien-1-one, 3,7-dimethyl-10-(1-methylethylidene)–1IYL complex (Figure 4B), the Lig fit Prot stayed significantly lower than the protein’s RMSD throughout the simulation, suggesting that the orientation of the ligand remained the same during the simulation process. For the 3,7-cyclodecadien-1-one, 3,7-dimethyl-10-(1-methylethylidene)–1HD2 complex (Figure 4C), the Lig fit Prot value stabilized up to 40 ns, suggesting the casuarinin changing posed after 40 ns and then stabilized to a constant pose, and for the 3,7-cyclodecadien-1-one, 3,7-dimethyl-10-(1-methylethylidene)–1HNY complex (Figure 4D), the Lig fit Prot value stayed significantly lower than the protein’s RMSD throughout the simulation, suggesting that the orientation of the ligand remained the same during the simulation process.

#### 2.6.2. RMSF of Protein–Ligand Complexes

The root mean square fluctuation (RMSF) is useful for portraying confined changes along the protein chain (Figure 5). In the graph, the peaks demonstrate regions of the protein that vary the most throughout the simulation. Ordinarily, the tails (N- and C-termini) show maximum change as compared to other internal regions of the protein. Secondary regions of proteins such as alpha helices and beta strands are generally more inflexible and rigid than the unstructured regions and hence vacillate, not exactly like loop-forming portions of protein. Alpha-helical and beta-strand areas are featured in red and blue foundations separately. These districts are characterized by helices or strands that endure over 70% of the whole re-enactment. Protein deposits that contact ligands are set apart by green-hued vertical bars. The RMSF of the protein can likewise be related to the exploratory x-beam B-factor (right Y-hub). Because of the distinction between the RMSF and B-factor definitions, balanced correspondence ought not to be normal. Notwithstanding, the reproduction results should resemble crystallographic information. It was found that the RMSF plot for 3,7-cyclodecadien-1-one, 3,7-dimethyl-10-(1-methylethylidene) fit over 1EA1 and 1IYL proteins and showed less residual fluctuation within the range of 0.8–1.6 Å in α-helical and β-strands (Figure 5A, B). The RMSF plot for 3,7-cyclodecadien-1-one, 3,7-dimethyl-10-(1-methylethylidene)–1HD2 was found to show less residual fluctuation in α-helical and β-strands between 0.6 and 1.6 Å (Figure 5C), while the 3,7-cyclodecadien-1-one, 3,7-dimethyl-10-(1-methylethylidene)–1HNY complex was found to be a fit over proteins with less fluctuation (Figure 5D). 

Protein interactions with the ligand can be monitored throughout the simulation. These interactions can be categorized by type and summarized, as shown in Figure 6. 3,7-cyclodecadien-1-one, 3,7-dimethyl-10-(1-methylethylidene) in complex with 1EA1 showed hydrogen bonding with Gln 72 and Arg 96; water bridges with Met 325, Arg 326, and Arg 393; and hydrophobic interactions with Tyr 76, Phe 78, Met 79, Phe 83, Met 99, Phe 255, Leu 321, Leu 324, Cys 394, Val 395, Met 433, and Val 434 (Figure 6A). 3,7-Cyclodecadien-1-one, 3,7-dimethyl-10-(1-methylethylidene) in complex with 1IYL showed hydrogen bonding with Asn 392; water bridges with Gly 413; and hydrophobic interactions with Phe 115, Phe 117, Tyr 225, Leu 235, Phe 240, Phe 339, Leu 350, Ile 352, and Val 390 (Figure 6B). 3,7-Cyclodecadien-1-one, 3,7-dimethyl-10-(1-methylethylidene) in complex with 1IHD2 showed hydrogen bonding with Thr 44 and Gly 46; water bridges with Lys 49, Thr 50, Glu 53, His 88, Arg 127, Asp 145, and Thr 147; and hydrophobic interactions with Pro 40, Phe 43, Pro 45, Pro 53, PHE 120, and Leu 149 (Figure 6C). 3,7-Cyclodecadien-1-one, 3,7-dimethyl-10-(1-methylethylidene) in complex with 1HNY showed hydrogen bonding with Asp 197; water bridges with Glu 233; and hydrophobic interactions with Trp 58, Trp 59, Tyr 62, Val 98, Leu 162, Leu 165, Ile 235, and Phe 256 (Figure 6D). The total number of specific contacts of the ligand with selected proteins was also studied throughout the simulation (0–100 ns). Some residues of proteins were found to show more than one specific contact with the ligands, which is represented by a darker shade of orange color, as shown in Figure 7A–G.

### 2.7. Binding Free Energy Evaluation

Binding energy calculation provides an insight into the ligand potential to strongly interact with the amino acids of a target protein. After simulation analysis of the best docked phytocompounds, 3,7-cyclodecadien-1-one, 3,7-dimethyl-10-(1-methylethylidene) with all the target proteins was performed using MM-GBSA by taking snapshots of the trajectory profiles developed on performing the 100 ns MD simulation. Table 6 predicts the MM/GBSA profile of 3,7-cyclodecadien-1-one, 3,7-dimethyl-10-(1-methylethylidene) with all selected proteins and shows effective binding of this ligand with target proteins. Binding energy calculation provides an insight into the ligand potential to strongly interact with the amino acids of the protein. The energy released (∆G_bind_) due to bond formation, or rather interaction of the ligand with protein, is in the form of binding energy and it determines the stability of any given protein–ligand complex. The free energy of a favorable reaction is negative. It was observed that 3,7-cyclodecadien-1-one, 3,7-dimethyl-10-(1-methylethylidene) showed negative ∆G_bind_ with all target proteins. Van der Waals interactions (∆G_vdW_) of 3,7-cyclodecadien-1-one with the selected target proteins were found to be between −13.85 and −25.67 kcal/mol, suggesting that 3,7-cyclodecadien-1-one, 3,7-dimethyl-10-(1-methylethylidene) tends to stay in the vicinity of the interacting amino amides of target proteins. Coulomb energy was found to be negative for all complexes, indicating poor potential energy of 3,7-cyclodecadien-1-one, 3,7-dimethyl-10-(1-methylethylidene) with all target proteins and suggesting better stability of protein–ligand complexes. In addition to the total energy, the contributions to the total energy from different components such as hydrogen bonding correction, lipophilic energy, and Van der Waals energy is provided in Table 6.

### 2.8. Drug Likeness Prediction and Toxicity Prediction of Major Phytocompounds of CLO

The drug likeness filters help in the early preclinical development by avoiding costly late step preclinical and clinical failure. 3,7-Cyclodecadien-1-one, 3,7-dimethyl-10-(1-methylethylidene) has high bioavailability because it does not violate Lipinski’s rule of five, as it has a molecular mass of below 500 Da, possessing high lipophilicity (log P < 5), hydrogen donors (<5), and hydrogen acceptors (<10). Moreover, when we calculated the TPSA for passive molecular transport through membranes, the result showed their values were 17 Å^2^, having low oral bioavailability (Table 7). The results of toxicity prediction showed that the compound 3,7-cyclodecadien-1-one, 3,7-dimethyl-10-(1-methylethylidene) did not show any hepatotoxicity, immunogenicity, carcinogenicity, or cytotoxicity. The rodent toxicity (LD_50_) value of the 3,7-cyclodecadien-1-one, 3,7-dimethyl-10-(1-methylethylidene) compound was 5000 mg/kg (Class 5), indicating safer utilization of using 3,7-cyclodecadien-1-one, 3,7-dimethyl-10-(1-methylethylidene) as a potential drug (Table 7).

## 3. Materials and Methods

### 3.1. Chemicals and Media

The chemicals such as 2,2-diphenyl-1-(2,4,6-trinitrophenyl) hydrazyl (DPPH), 2,4,6-tri(2-pyridyl)-s-triazine (TPTZ), and *L-Ascorbic acid* were obtained from Sigma-Aldrich Co. LLC, Mumbai, India. Methanol and dimethyl sulphoxide (DMSO) were procured from Loba Chemie Pvt. Ltd., Mumbai, India. Alpha-amylase from *Aspergillus oryzae* and dinitrosalicylic acid (DNS) were purchased from Sigma-Aldrich Co. LLC, Mumbai. Other chemicals and reagents were of analytical grade, and the water used was double distilled. The media such as yeast peptone dextrose agar (YPDA) and yeast peptone dextrose broth (YPDB) were obtained from Himedia Laboratories Pvt. Ltd., Mumbai, India.

### 3.2. Collection and Identification of Plant Samples

The fresh leaves of *C. longa* were collected from Kangra, Himachal Pradesh, India (32.0998° N, 76.2691° E), in the month of October 2019. The plant specimen was identified in the Department of Forest Products at the Y.S. Parmar University of Horticulture and Forestry, Nauni, Solan, H.P., India. A sample voucher was submitted in the herbarium with voucher number UHF-965. The plant name was checked with the plant list (http://www.theplantlist.org, accessed on 7 October 2022).

### 3.3. Extraction of Essential Oil

Extraction of essential oil from *C. longa* leaves (CLO) was carried out by the hydro-distillation method using Clevenger assembly for oil lighter than water [53]. The leaves of *C. longa* were collected and thoroughly washed with distilled water to remove the dust particles, and then excess moisture was absorbed using a paper towel. About 200 g leaves of *C. longa* were cut into small pieces, mixed with distilled water, and boiled at 50 °C for 4 h in a round-bottom flask. Percentage extraction yield of CLO was determined on the basis of the weight of leaves and oil obtained. The collected CLO was stored at 4 °C in the dark for further analysis.

### 3.4. Evaluation of Antifungal Potential of CLO

#### 3.4.1. Fungal Strains and Growth Conditions

The two fungal strains *Candida albicans* (MTCC277) and *C. albicans* (MTCC90028) used in this study were procured from Microbial Type Culture Collection, Institute of Microbial Technology (IMTECH), Chandigarh, India. Both of these strains were maintained on potato dextrose agar (PDA) medium and grown in potato dextrose broth (PDB) at 28 ± 2 °C.

#### 3.4.2. Agar Well Diffusion Method for Antifungal Activity

Antimicrobial activity of CLO was determined using the agar well diffusion method [54] and was expressed as diameter of zone of inhibition (ZOI) against the tested strains. Fluconazole (Himedia Biosciences, Mumbai, India) was used as a positive control, whereas DMSO (solvent) was used as a negative control. The experiment was repeated twice, and results are expressed as mean ± S.D. 

#### 3.4.3. Minimum Inhibitory Concentration (MIC) of CLO Using the Micro Dilution Method

The MIC of CLO against tested fungal strains was determined using the micro dilution method according to the Clinical and Laboratory Standards Institute (CLSI) protocol [55]. The experiment was performed in a 96-well microtiter plate, and geometric dilutions (50–0.098 µg/mL) of CLO were prepared. Then, equal numbers of fungal cells (2 × 10^5^ CFU mL^−1^, 0.5 McFarland) were inoculated to each well, and the plate was incubated for 48 h at 28 ± 2 °C. Fluconazole was used as the positive control, and DMSO was used as the negative control. After incubation, resazurin dye (1 mg/mL) was added, and a change in color of resazurin dye was observed in each well. The lowest concentration at which color changed from purple to pink was considered as the MIC value. 

### 3.5. Analysis of Antioxidant Potential of CLO

The antioxidant capacity of CLO was evaluated using two different antioxidant assays, namely, DPPH and FRAP assays. For both assays, L-ascorbic acid (2.5–10 µg/mL) was used as the standard control [56,57,58]. The antioxidant capacity of CLO and ascorbic acid was expressed in terms of IC_50_ value (*half maximal* inhibitory concentration).

#### 3.5.1. DPPH Radical Scavenging Activity

The DPPH radical scavenging activity of CLO was measured by the method described by Torres-Martínez et al. [59]. In this procedure, 100 µL of CLO or ascorbic acid (10–80 µg/mL) was mixed with 900 µL of 0.004% DPPH solution, and the absorbance of the reaction mixture was measured at 517 nm after incubation of 30 min in the dark at room temperature using an ultraviolet–visible (UV–VIS) spectrophotometer. The capability of scavenging the DPPH radical was calculated from the following equation:$$\%\;\mathrm{DPPH}\;\mathrm{radical}\;\mathrm{scavenging}\;\mathrm{activity}=\frac{A\;\left(\mathrm{control}\right)-A\;\left(\mathrm{sample}\right)}{A\;\left(\mathrm{control}\right)}\times 100$$
where A (control) is the absorbance of the control, and A (sample) is the absorbance of the test/standard.

#### 3.5.2. FRAP Assay

The FRAP activity of CLO was expressed as Fe (II) equivalents per gram of the extract calculated from the linear calibration curve of FeSO_4_ (10 to 80 μM) as described by Kumar et al. [54] and Kumar et al. [58]. To 100 µL of CEO or ascorbic acid (10–80 µg/mL), 900 µL of freshly prepared FRAP solution was added. The FRAP reagent was prepared by mixing 300 mM acetate buffer (pH-3.6), 10 mM TPTZ in 40 mM HCl, and 20 mM FeCl_3_ at a ratio of 10:1:1 (*v*/*v*/*v*). The reaction mixture was incubated at room temperature for 30 min, and then absorbance was recorded at 593 nm using a UV–VIS spectrophotometer. 

### 3.6. Evaluation of Anti-Diabetic Potential of CLO

The anti-diabetic potential of CLO was evaluated using an in vitro α-amylase inhibition method. In this method, the enzyme solution was prepared by dissolving α-amylase in 20 mM phosphate buffer (pH-6.9) at a concentration of 0.5 mg/mL. Then, 1 mL of CLO of various concentrations (10–80 µg/mL) was mixed with 1 mL of enzyme solution and incubated at 25°C for 10 min. After incubation, 1 mL of starch (0.5%) solution was added to the mixture, and further reaction mixture was incubated at 25°C for 10 min. The reaction was terminated by adding 2 mL of dinitrosalicylic acid (DNS, color reagent) and heating the reaction mixture in a boiling water bath for 5 min. Then, absorbance was measured after cooling at 540 nm [60,61]. Metformin was used as the standard drug in this experiment. The inhibition percentage was calculated using the following formula:$$\mathrm{Percentage}\;\mathrm{inhibition}=\frac{A\;\left(\mathrm{control}\right)-A\;\left(\mathrm{sample}\right)}{A\;\left(\mathrm{control}\right)}\times 100$$
where A (control) is the absorbance of the control reaction (containing all reagents except the test sample) and A (sample) is the absorbance of the test sample. The experiment was performed in triplicate, and results were calculated as mean ± S.D.

### 3.7. Identification of Chemical Components of CLO using GC–MS Analysis

The chemical composition of CLO was conducted using the GC–MS technique using Thermo Trace 1300 GC coupled with a Thermo TSQ 800 Triple Quadrupole mass spectrometer fitted with a BP 5MS capillary column (30 m 0.25 mm, 0.25 mm film thickness). Helium was used as the carrier gas at a flow rate of 1 mL/min. The oven program started with an initial temperature of 50°C and was then held for 5 min; following this, the temperature was heated at rate of 5°C/min to 280°C and finally held isothermally for 2 min. The run time was 34.09 min. The MS operated at a flow speed of 1 mL/min, with an ionization voltage of 70 eV, at an interface temperature of 250°C, in a SCAN mode, and at a mass interval of m/z 35–650. The essential oil constituents were identified in relation to the reference on the basis of their retention time (R_T_). The compounds were identified on the basis of matching unknown peaks with the MS data bank (NIST 2.0 Electronic Library).

### 3.8. Molecular Docking of Major Constituents of CLO with Antifungal, Antioxidant, and Anti-Diabetic Protein Targets

#### 3.8.1. Ligand Preparation

The 3-D structure phytocompounds were obtained from the Pubchem database (https://pubchem.ncbi.nlm.nih.gov/, accessed on 5 October 2022) in sdf format and were energy minimized using the Chem3D structure software and saved as pdb files. The selected ligands were (-)-zingiberene; 3,7-cyclodecadien-1-one, 3,7-dimethyl-10-(1-methylethylidene); cyclohexene, 4-methyl-3-(1-methylethylidene); and (+)-4-carene (Appendix A).

#### 3.8.2. Retrieval and Preparation of Target Proteins

Two antifungal targets, namely, N-myristoyl transferase (NMT; PDB ID: 1IYL) of *C. albicans* [62] and cytochrome P_450_ 14 alpha–sterol demethylase (CYP51; PDB ID: 1EA1) of *Mycobacterium tuberculosis* [63]; one antioxidant target, human peroxiredoxin 5 (PDB ID: 1HD2) [64]; and one anti-diabetic target, human pancreatic alpha-amylase [65] (PDB ID: 1HNY) were selected for studying the *in-silico* interaction of phytocompounds of CLO. The 3-D crystal structures of selected target proteins were obtained from the RCSB protein data bank (http://www.rcsb.org/, accessed on 5 October 2022) (Appendix A).

The crystal structures of target proteins were prepared for binding analysis using Autodock Tools (ADT). Protein preparation included the addition of Gasteinger charges, polar hydrogen atoms, and optimizing the rotatable bonds. Prepared proteins were then saved in pdbqt format for further analysis. Further, binding sites of target proteins were obtained from the previous literature, and the grid box was created on the basis of the above information [53,66,67,68,69]. The details of target proteins, number of amino acids, chain selected for docking, and grid box coordinates are shown in Table 8.

#### 3.8.3. Molecular Docking

Molecular docking of phytocompounds with selected proteins was performed using the Glide (grid-based ligand docking) program incorporated in the Schrödinger molecular modelling package with extra precision (XP). Extra-precision (XP) docking and scoring is a more powerful and discriminating procedure that requires more time to execute than SP. XP is intended for use on ligand postures that have been demonstrated to be high-scoring using standard precision (SP) docking. XP also has a more complicated scoring methodology that is “harder” than the SP GlideScore, with stricter ligand–receptor form complementarity criteria. This eliminates false positives that SP allows through. Because XP penalizes ligands that do not match well to the specific receptor conformation used, we recommend docking to many receptor conformations whenever possible. The best pose based on binding energies for each ligand–protein interaction was further analyzed in Discovery Studio (DS) Visualizer (Accelrys, San Diego, CA, USA). From the interaction profile, the ligands showing high binding energy were further considered for the molecular dynamic simulations.

#### 3.8.4. MD Simulations

Structural stability of the receptor–ligand complexes was investigated using MD simulations with the help of the academic version of the Desmond program [70,71]. For this, the system was designed using the TIP3P water model with a cubic periodic box containing simple point charge (SPC) (10 Å × 10 Å × 10 Å) and optimized potentials for liquid simulations (OPLS) all-atom force field 2005 [72]. Then, the appropriate amount of sodium ions was added for the system neutralization process. The receptor–ligand complex was provided for the initial energy minimization step and pre-equilibration in various restrained steps. 

MD simulations were carried out using OPLS 2005 force field parameters with periodic boundary conditions in the NPT ensemble system [73,74], with a relaxation duration of 1 ps at a constant temperature of 300 K and a constant volume. The smooth particle mesh Ewald (PME) approach (with a 10^−9^ tolerance limit and a cut off distance of 9.0 Å) was used to analyze protein structures every 1 ns. An average structure from the MD simulation corresponding to the production period was used to determine the stability. Furthermore, the root means square deviation (RMSD), the root means square fluctuation (RMSF), the hydrogen bond, the radius of gyration (R_g_), and the histogram for torsional bonds were investigated for the analysis of structural changes with the dynamic role of the receptor–ligand complexes [75,76,77].

#### 3.8.5. MM-GBSA (Molecular Mechanics Generalized Born Surface Area) Binding Energies

MM-GBSA and molecular mechanics Poisson–Boltzmann surface area (MM-PBSA) were employed for the calculation of the binding free energies of protein–ligand complexes [78,79]. Hence, the PRIME module of Maestro 11.4 and the OPLS-2005 force field were used for the determination of the binding energy of best-docked ligand–receptor complex, and the following equation was used for the calculation of binding energy:∆G_Bind_ = ∆E_MM_ + ∆G_Solv_ + ∆G_SA_

where ∆E_MM_ represents the difference of the minimized energies of the protein–ligand complex, while ∆G_Solv_ is the difference between GBSA solvation energy of the protein–ligand complexes and the sum of the solvation energies for the protein and ligand. ∆G_SA_ represents the surface area energies in the protein–ligand complexes and the difference in the surface area energies for the complexes [80,81]. The protein–ligand complexes were minimized using a local optimization feature of PRIME.

### 3.9. Drug Likeness, ADME/Toxicity Prediction

Lipinski’s rule (rule of five, RO5) was considered the primary factor for screening of the molecules, and it was evaluated using the SWISS ADME web server (http://www.swissadme.ch/, accessed on 6 October 2022). Further, the toxicity of selected compounds was analyzed using the Protox-II tool to ascertain their risk of drugability [82]. PROTOX is a server that predicts the LD_50_ value and toxicity class of a question molecule in rodents. The SMILES format of the selected compounds was submitted to a Swiss ADME web server and Protox-II tool.

### 3.10. Statistical Analysis

The results are represented as mean ± standard deviation (SD) wherever applicable. The statistical comparisons were conducted using two-way analysis of variance (ANOVA) (*p* < 0.05) using Graph Pad Prism 8.0 (GraphPad Software, San Diego, CA-92108, USA).

## 4. Conclusions

Traditional medicinal herbs offer a wealth of phytocompounds, including essential oils (EOs), which can be explored for antifungal activities. Essential oil of *C. longa* (CLO) leaves showed antifungal, antioxidant, and anti-diabetic activity that was further validated by in silico studies. Among selected phytocompounds, 3,7-cyclodecadien-1-one, 3,7-dimethyl-10-(1-methylethylidene) of CLO showed higher interaction towards the antifungal, antioxidant, and anti-diabetic receptors, which was further validated with MD simulations. Further, 3,7-Cyclodecadien-1-one, 3,7-dimethyl-10-(1-methylethylidene) was found to be safer for drug formulation as it follows Lipinski’s rule and lacks hepatotoxicity, immunogenicity, carcinogenicity, and cytotoxicity. In light of these findings, we can say that the essential oil of *C. longa* (CLO) leaves can be exploited for its broad-spectrum therapeutic applications.

## Figures and Tables

**Figure 1 molecules-27-07664-f001:**
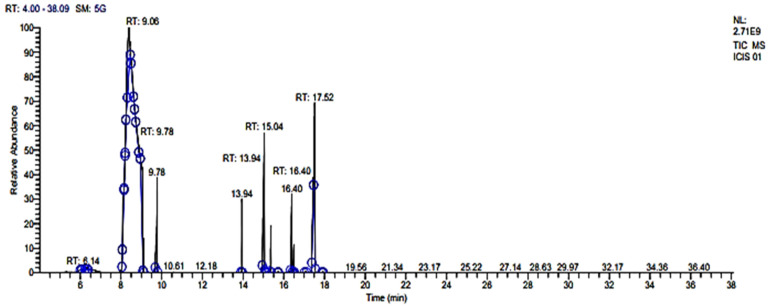
Chemical composition of CLO through GC–MS. Phytoconstituents were identified through different time interval and recorded in terms of retention time (RT).

**Figure 2 molecules-27-07664-f002:**
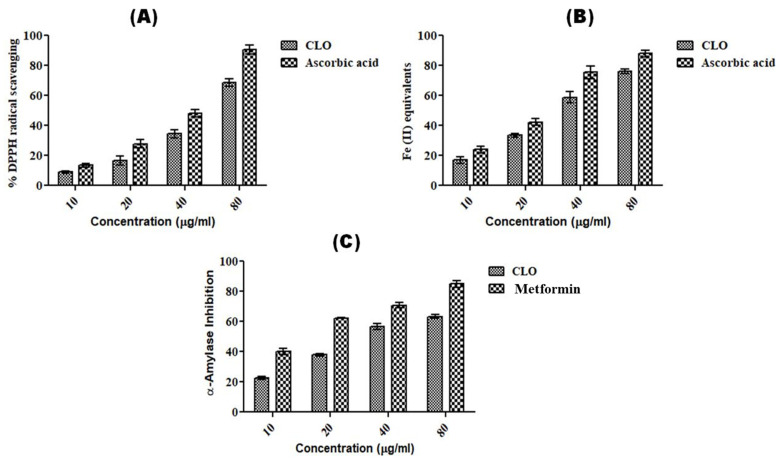
Dose-dependent antioxidant and anti-diabetic activity shown by CLO. (**A**) % DPPH radical scavenging method; (**B**) FRAP method; (**C**) α-amylase inhibition assay. Ascorbic acid and metformin were used as standard controls in antioxidant and anti-diabetic assays. The experiments were repeated thrice, and values are expressed as mean ± S.D. (n = 3).

**Figure 3 molecules-27-07664-f003:**
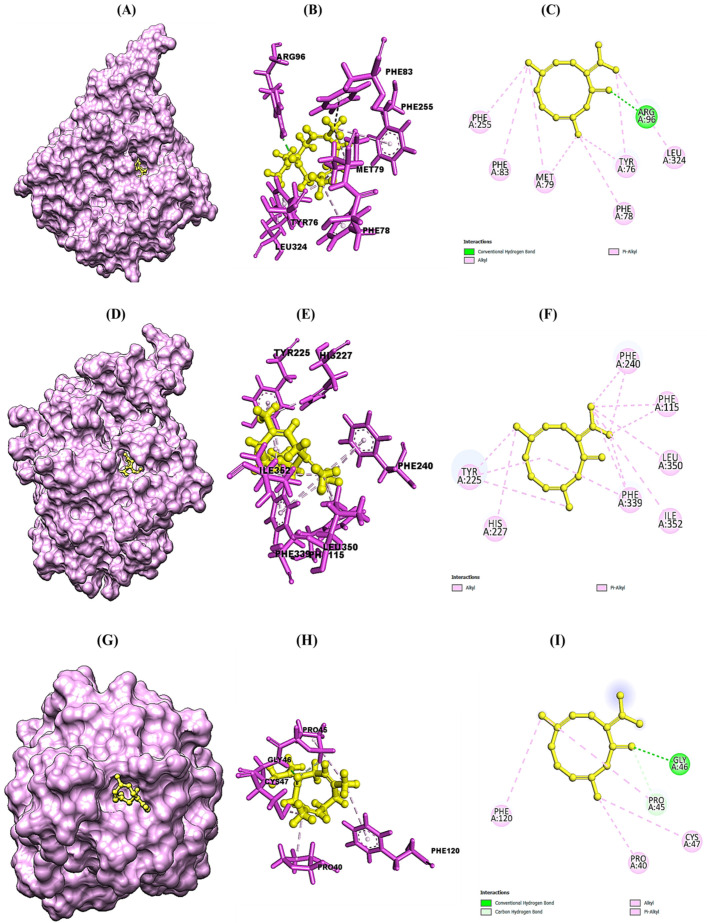
Structural representation of molecular docking analysis of 3,7-cyclodecadien-1-one, 3,7-dimethyl-10-(1-methylethylidene) with target proteins. (**A**,**D**,**G**,**J**) Binding of ligand inside 1EA1, 1IYL, IHD2, and IHNY proteins, respectively; (**B**,**E**,**H**,**K**) 3-D interactions of the ligand with the interacting amino acids of the selected 1EA1, 1IYL, IHD2, and IHNY proteins, respectively; (**C**,**F**,**I**,**L)** 2-D interactions of the ligand with interacting amino acids of the selected 1EA1, 1IYL, IHD2, and IHNY proteins, respectively. Interactions were analyzed using Discovery Studio 2021 client software. Different colors indicate different types of interactions, namely, Van der Waals interactions in light green, conventional hydrogen bonds in green color, π–sigma in purple color, π–π–T shaped in dark pink color, and alkyl and π–alkyl bonds in light pink color.

**Figure 4 molecules-27-07664-f004:**
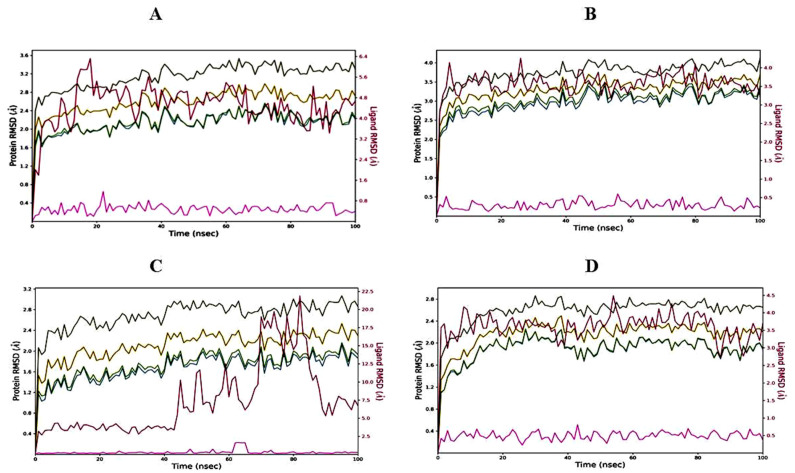
RMSD graph of 3,7-cyclodecadien-1-one, 3,7-dimethyl-10-(1-methylethylidene) of CLO with target proteins. (**A**) 3,7-cyclodecadien-1-one, 3,7-dimethyl-10-(1-methylethylidene) in complex with 1EA1; (**B**) 3,7-cyclodecadien-1-one, 3,7-dimethyl-10-(1-methylethylidene) in complex with 1IYL; (**C**) 3,7-cyclodecadien-1-one, 3,7-dimethyl-10-(1-methylethylidene) in complex with 1HD2; (**D**) 3,7-cyclodecadien-1-one, 3,7-dimethyl-10-(1-methylethylidene) in complex with 1HNY protein. Color legends: Ca (blue color), side chains (green color), heavy atoms (yellow color), ligand with protein (dark pink color), ligand with ligand (pink color).

**Figure 5 molecules-27-07664-f005:**
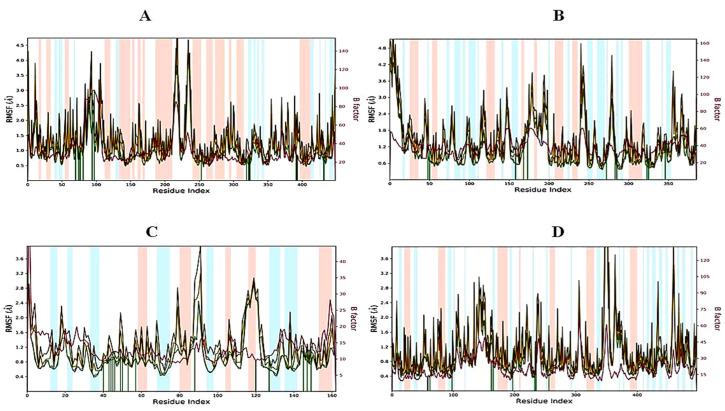
RMSF graph of 3,7-cyclodecadien-1-one of CLO with target proteins for 100 ns: (**A**) 3,7-cyclodecadien-1-one in complex with 1EA1, (**B**) 3,7-cyclodecadien-1-one in complex with 1IYL, (**C**) 3,7-cyclodecadien-1-one in complex with 1HD2, (**D**) 3,7-cyclodecadien-1-one in complex with 1HNY protein. Color legends: Cα (blue color), side chains (green color), heavy atoms (yellow color), ligand with protein (dark pink color), ligand with ligand (pink color).

**Figure 6 molecules-27-07664-f006:**
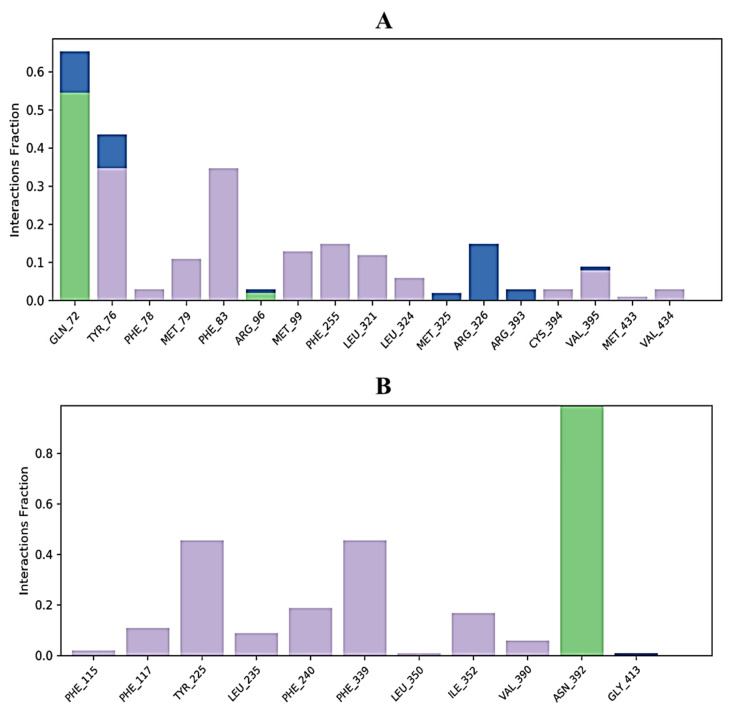
Histogram of ligand contacts with amino acid residues of target proteins. (**A**) 3,7-Cyclodecadien-1-one, 3,7-dimethyl-10-(1-methylethylidene) in complex with 1EA1; (**B**) 3,7-cyclodecadien-1-one, 3,7-dimethyl-10-(1-methylethylidene) in complex with 1IYL; (**C**): 3,7-cyclodecadien-1-one, 3,7-dimethyl-10-(1-methylethylidene) in complex with 1HD2; (**D**) 3,7-cyclodecadien-1-one, 3,7-dimethyl-10-(1-methylethylidene) in complex with 1HNY protein. Different types of bar color indicate different types of bonds: hydrogen bond (green), hydrophobic contacts (purple), and water-bridge (blue).

**Figure 7 molecules-27-07664-f007:**
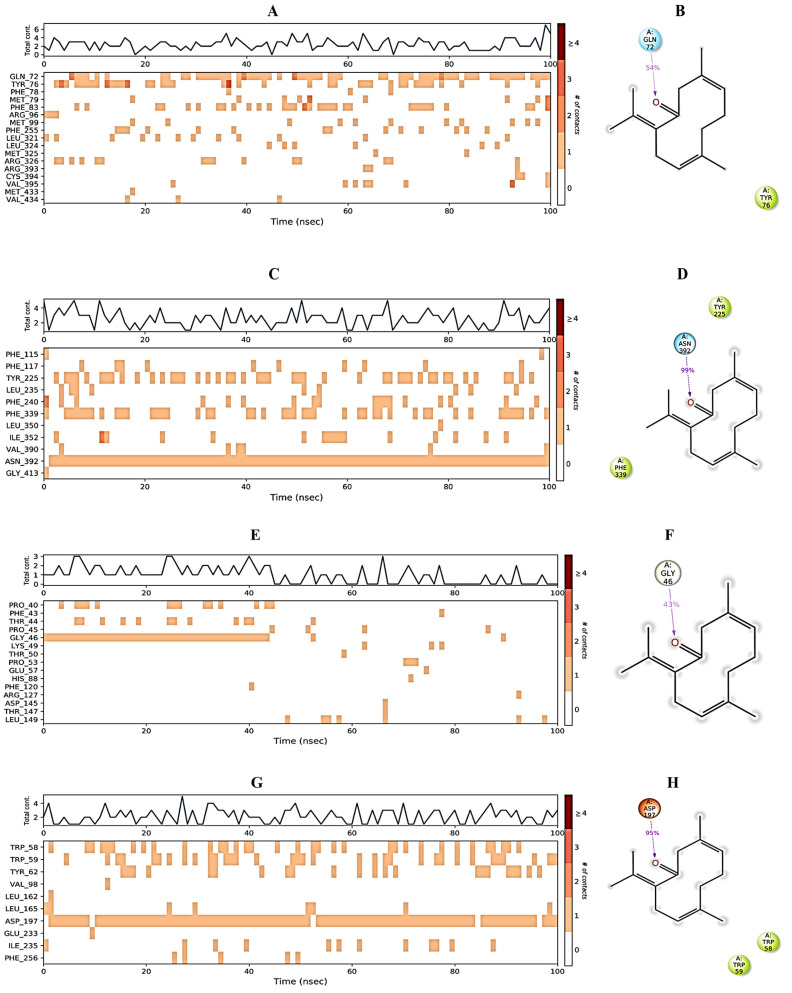
Protein–ligand contact. (**A**,**C**,**E**,**G**) The total number of specific protein–ligand contacts over the course of the trajectory. (**B**,**D**,**F**,**H**) Residues of 1EA1, 1IYL, 1HD2, and 1HNY proteins interacting with the ligand in each trajectory frame, respectively.

**Table 1 molecules-27-07664-t001:** Major compounds identified in CLO identified through GC–MS.

Compound Name	RT (min)	Area %	Molecular Weight (g/mol)	Molecular Formula	RI_C_	RI_L_
3-Decen-1-yne, (E)-	6.06	0.10	136.24	C_10_H_16_	956	-
1,11-Dodecadiyne	6.14	0.26	162.27	C_12_H_18_	1011	1012
2-Nonynoic acid	6.30	0.13	154.21	C_9_H_14_O_2_	1028	-
2-Carene	8.08	3.28	136.24	C_10_H_16_	1070	1001
Terpinolene	8.30	5.97	136.24	C_10_H_16_	1088	1078
(+)-4-Carene	8.37	11.89	136.24	C_10_H_16_	1015	1120
Cyclohexene,4-methyl-3-(1-methylidene)-	9.06	12.47	108.18	C_8_H_12_	1120	1125
1,6-Octadien-3-ol, 3,7-dimethyl	9.11	1.67	196.29	C_10_H_18_O	1125	1130
Camphor	9.78	8.73	152.23	C_10_H_16_O	1139	1148
Caryophyllene	13.94	2.78	204.35	C_15_H_24_	1440	1420
(-)-Zingiberene	15.04	17.84	204.35	C_15_H_24_	1458	1460
cis-α-Farnesene	15.37	2.66	204.35	C_15_H_24_	1465	1470
α-Elemenone	16.40	6.38	218.33	C_15_H_22_O	1468	1475
cis-Sesquisabinene hydrate	16.50	2.21	222.37	C_15_H_26_O	1470	1477
cis-β-Elemenone	17.44	3.39	190.28	C_15_H_22_O	1592	1589
Germacrone	17.52	15.31	218.33	C_15_H_22_O	1690	1693

RT—retention time in minutes, Area %—percentage area, RI_C_—calculated retention index, RI_L_—retention index reported from previous reports.

**Table 2 molecules-27-07664-t002:** Antifungal activity exhibited by CLO using agar well diffusion and the broth dilution method.

Fungal Strains	ZOI (mm)			
	*C. albicans* (MTCC90028)		*C. albicans* (ATCC277)	
Volume used	25 µL	50 µL	25 µL	50 µL
CLO	12.5 ± 0.71	13 ± 1.41	10.5 ± 0.71	11.5 ± 0.71
Fluconazole *	18 ± 0.7		13 ± 0.71	
**MIC (%)**				
CLO	0.63 ± 0		0.63 ± 0	
Fluconazole *	0.063 ± 0		0.063 ± 0	

* Fluconazole was used as a positive control in both experiments. Values are expressed as mean ± S.D. of two independent experiments.

**Table 3 molecules-27-07664-t003:** Half maximal inhibitory concentration (IC_50_) of CLO, ascorbic acid, and metformin in terms of antioxidant assay and anti-diabetic assay. DPPH and α-amylase inhibition activity were calculated in terms of µg/mL, while FRAP activity was calculated in terms of µM Fe (II) equivalents. The lower the value of IC_50_, the higher the antioxidant/anti-diabetic potential.

EO/Standard Drugs	Antioxidant Activity		Anti-Diabetic Activity
	DPPH Assay	FRAP Assay	α-Amylase Inhibition
**CLO**	5.85 ± 1.61 ^a^	32.92 ± 0.64 ^a^	43.06 ± 2.51 ^a^
**Ascorbic acid**	3.11 ± 0.47 ^ab^	24.09 ± 2.16 ^ab^	-
**Metformin**	-	-	16.51 ± 2.11 ^ab^

Different superscripts (a—CLO; b—positive control) on data value show significant (*p* < 0.0001) variation in biological activities of CLO with respect to positive control (two-way ANOVA). The values are expressed as mean ± S.D. (n = 3).

**Table 4 molecules-27-07664-t004:** Docking scores of phytocompounds and standard drugs with target protein receptors.

Phytocompounds/Standard Drugs	Docking Scores (kcal mol^−1^)				Glide Energy (kcal mol^−1^)			
	1IYL	1EA1	1HNY	1HD2	1IYL	1EA1	1HNY	1HD2
3,7-Cyclodecadien-1-one, 3,7-dimethyl-10-(1-methylethylidene)	−6.697	−4.978	−2.708	−2.447	−24.223	−21.331	−20.819	−19.399
Cyclohexene, 4-methyl-3-(1-methylethylidene)	−5.961	−3.883	−1.661	−1.707	−22.224	−17.617	−15.129	−13.515
(+)-4-Carene	−5.317	−5.27	−1.641	−1.94	−17.873	−18.074	−17.275	−14.839
(-)-Zingiberene	−3.934	−4.794	−1.229	−2.141	−14.071	−14.282	−10.362	−10.914
Fluconazole	−7.716	−6.516	-	-	−38.248	−37.349	-	-
Metformin	-	-	−2.972	-	-	-	−17.117	-
Ascorbic acid	-	-	-	−6.981	-	-	-	−23.999

**Table 5 molecules-27-07664-t005:** Interacting amino acids of target proteins with selected phytocompounds and drug candidates.

Phytocompounds/ Standard Drugs	Interacting Amino Acids			
	1IYL	1EA1	1HNY	1HD2
3,7-Cyclodecadien-1-one	PHE:115, TYR:225, HIS:227, PHE:240, PHE:339, LEU:350, ILE:352	TYR:76, PHE:78, MET:79, PHE:83, ARG:96, PHE:255, LEU:324	TRP:58, TRP:59, HIS:101, LEU:165, ALA:198, HIS:305	PRO:40, PRO:45, GLY:46, CYS:47, PHE:120
Cyclohexene, 4-methyl-3-(1-methylethylidene)	TYR:225, TYR:354, LEU:394, LEU:415,	PHE:255, MET:79, PHE:78, LEU:321, TYR:76	TYR:151, LEU:162, ALA:198, LYS:200, HIS:201, ILE:235	PRO:45, PRO:40, LEU:116, ILE:119, PHE:120
(+)-4-Carene	PHE:115, TYR:225, HIS:227, PHE:240, PHE:339, TYR:354	TYR:76, PHE:78, MET:79, PHE:255, HIS:259, LEU:321, VAL:434	TYR:62, HIS:101, LEU:162, LEU:165, ALA:198	PRO:40, PRO:45, CYS:47, LEU:116, PHE:120
(-)-Zingiberene	TYR:354, LEU:394	TYR:76, PHE:78, LEU:321	ALA:198, LYS:200, HIS:201, ILE:235	PRO:40, PHE:120
Fluconazole	PHE:115, TYR:225, HIS:227, PHE:240, TYR:354, ASN:392	TYR:76, MET:79, PHE:83, ARG:96, MET:99, LEU:100, SER:252, PHE:255, ALA:256, HIS:259, LEU:321	-	-
Metformin	-	-	TYR:62, ARG:195, ASP:197, GLU:233, ASP:300	-
Ascorbic acid	-	-		PRO:45, GLY:46, CYS:47, ARG:127

**Table 6 molecules-27-07664-t006:** MM/GBSA profiles of 3,7-cyclodecadien-1-one while interacting with target proteins.

Protein–Ligand Complex	∆G_Bind_	∆G_Coulomb_	∆G_vdW_	∆G_Ligand_efficiency_
1EA1_3,7-Cyclodecadien-1-one, 3,7-dimethyl-10-(1-methylethylidene)	−25.80	−5.23	−21.02	−1.58
1IYL_3,7-Cyclodecadien-1-one, 3,7-dimethyl-10-(1-methylethylidene)	−37.11	−9.21	−25.67	−2.27
1HD2_3,7-Cyclodecadien-1-one, 3,7-dimethyl-10-(1-methylethylidene)	−18.69	−6.79	−13.85	−1.13
1HNY_3,7-Cyclodecadien-1-one, 3,7-dimethyl-10-(1-methylethylidene)	−26.99	−9.59	−21.97	−1.63

Coulomb—Coulomb energy. H-bond—hydrogen bonding correction. Lipo—lipophilic energy. vdW—Van der Waals energy.

**Table 7 molecules-27-07664-t007:** Drug likeness prediction and toxicity prediction of the best docked ligand (3,7-cyclodecadien-1-one, 3,7-dimethyl-10-(1-methylethylidene) using the Protox-II server.

Properties	3,7-Cyclodecadien-1-one, 3,7-dimethyl-10-(1-methylethylidene)
Log P	4.36
TPSA	17.07
MW	218.34
Number of acceptor H and O	1
Number of donor H and O	0
Violations	0
Lipinski rule	Yes
Hepatotoxicity	No
Immunogenicity	No
Carcinogenicity	No
Cytotoxicity	No
LD_50_	5000 mg/kg (Class 5)

Log P—measure of molecular hydrophobicity; TPSA—topological polar surface area; MW—molecular weight; LD_50_—lethal dose.

**Table 8 molecules-27-07664-t008:** Details of target proteins and grid box coordinates for docking.

Target Proteins	Amino Acids	Resolution	Chain Selected for Docking	Grid Box Co-Ordinates
N-Myristoyl transferase (NMT; PDB ID: 1IYL)	392	3.20 Å	Chain-A	x = 11.256; y = 49.911; z = 1.04 (X = 40; Y = 40; Z = 40)
Cytochrome P_450_ 14 alpha–sterol demethylase (CYP51; PDB ID: 1EA1)	449	2.21 Å	Chain-A	x = 17.702; y = −3.978; z = 67.221 (X = 40; Y = 40; Z = 40)
Human peroxiredoxin 5 (PDB ID: 1HD2)	161	1.50 Å	Chain-A	x = 7.44; y = 41.368; z = 38.078 (X = 54; Y = 40; Z = 40)
Human pancreatic alpha-amylase (PDB ID: 1HNY)	496	1.80 Å	Chain-A	x = 13.84; y = 45.519; z = 16.581 (X = 68; Y = 66; Z = 60)

## Data Availability

Not applicable.

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
