# Peer review of "Evaluation of the Antifungal, Antioxidant, and Anti-Diabetic Potential of the Essential Oil of Curcuma longa Leaves from the North-Western Himalayas by In Vitro and In Silico Analysis"

_molecules, 2022, doi:10.3390/molecules27227664_

Round 1
Reviewer 1 Report
Authors report an in-vitro and in-silico study to evaluate the antifungal, antioxidant, and anti-diabetic potential of essential oil extracted from Curcuma longa leaves.
The study is well structured, developed and the results are critically discussed also in the light of the present literature.
The authors reported an accurate experimental approach. Figures and tables help the reader to interpret the results and the discussion
My opinion is that the manuscript can be accepted in its present form
Author Response
Thanks for your kind comments. We have checked the manuscript for language and style.
Reviewer 2 Report
The study aimed to explore the phytochemicals as well as antifungal, antioxidant, and anti-diabetic potential pharmaological properties of essential oil of Curcuma longa leaves from North-western Himalayas by in vitro and in silico analysis.
Findings demonstrate that essential oil of C. longa (CLO) leaves can be exploited for its broadspectrum therapeutic applications.
The methods used are sufficiently documented and allow replication studies. Results obtained are well explained and data interpretation is also correct. Conclusions are consistent with the evidence and arguments presented.
About limitations the authors should include statistical analysis. About strenghts the authors explored the topic and they obtained the purpose of the study.
However, I will recommend the acceptance of this manuscript after these modifications:
1. The introduction is definitely too short - please expand this part and emphasize the novelty of the work;
2. The reported acronymus should be explain in extenso at their first appearance in the text: MIC, GC-MS, MM-GBSA, ZOI, DPPH, FRAP, ..;
3. In Materials and Methods the section on statistical analysis is completely missing;
4. In the text and In the legend of Figure 2 and Tables 2,3 the authors should add informations about statistical analysis;
5. In Figure 2 where is metformin as positive control? In graph A, B or C?
6. About references: authors should substitute if they can following too old bibliographic references 52, 59 and 65.
Author Response
The study aimed to explore the phytochemicals as well as antifungal, antioxidant, and anti-diabetic potential pharmacological properties of essential oil of Curcuma longa leaves from North-western Himalayas by in vitro and in silico analysis.
Findings demonstrate that essential oil of C. longa (CLO) leaves can be exploited for its broad-spectrum therapeutic applications.
The methods used are sufficiently documented and allow replication studies. Results obtained are well explained and data interpretation is also correct. Conclusions are consistent with the evidence and arguments presented.
About limitations the authors should include statistical analysis. About strengths the authors explored the topic and they obtained the purpose of the study.
However, I will recommend the acceptance of this manuscript after these modifications:
- The introduction is definitely too short - please expand this part and emphasize the novelty of the work;
Reply: Thanks for your comments. The introduction part has been expanded as suggested by the reviewer.
- The reported acronymus should be explain in extenso at their first appearance in the text: MIC, GC-MS, MM-GBSA, ZOI, DPPH, FRAP, ..;
Reply: Thanks for your comments. As per suggestion of the reviewer, the acronyms are expanded in at their first appearance.
- In Materials and Methods the section on statistical analysis is completely missing;
Reply: Thanks for your comments. The statistical analysis has been added in materials and methodology section as suggested by the reviewer.
- In the text and In the legend of Figure 2 and Tables 2,3 the authors should add informations about statistical analysis;
Reply: Thanks for your comments. The information about statistical analysis has been added as suggested by the reviewer.
- In Figure 2 where is metformin as positive control? In graph A, B or C?
Reply: Thanks for your comments. The information about metformin has been corrected in Fig 2C as suggested by the reviewer.
- About references: authors should substitute if they can follow too old bibliographic references 52, 59 and 65.
Reply: Thanks for your comments. As suggested by the reviewers, we have replace old references with new references.
Reviewer 3 Report
My comments on the manuscript entitled "Evaluation of antifungal, antioxidant, and anti-diabetic potential of essential oil of Curcuma longa leaves from North-western Himalayas by in vitro and in silico analysis" is as follows;
1. Line 54-55: check the sentence as gives an idea that turmeric leaves are discarded as it has high nutritive values.
2. The introduction needs to be improved with background information on fungal diseases, free radical and problems related to it (with a focus on carcinogenesis) and diabetes as the manuscript focuses on these areas. Also I suggest to start either by describing about "essential oils" or "Zingiberaceae" rather than a generalized description on natural products.
3. Why the entire compound from GC-MS not listed out? It is always better to include all the chemicals. In addition, the table should be modified by including the RI/ KI values obtained during experiment as well as based on reference (NIST library).
4. There are several reports on antifungal and antioxidant properties of essential oil from Curcuma longa leaves. What is the significance in repeating the same in the present study?
5. The authors haven’t mentioned about the statistical tools used in validating the significance of the results,
6. All scientific names should be in italics (Starting from Keywords there were several mistakes)
7. The anti-diabetic activity can not be taken as a strong indication, especially based on a single assay. I suggest to inlcude the inhibitory potential of alpha glucosidase also.
8. Though the authors used a common "Results and Discussion", they failed to emphasize the discussion part. Mentioning previous studies don't make the discussion, authors needs to compare the existing results with those previous reports. I suggest to thoroughly modify this section.
Author Response
Reviewer-3
My comments on the manuscript entitled "Evaluation of antifungal, antioxidant, and anti-diabetic potential of essential oil of Curcuma longa leaves from North-western Himalayas by in vitro and in silico analysis" is as follows;
- Line 54-55: check the sentence as gives an idea that turmeric leaves are discarded as it has high nutritive values.
Reply: Thanks for your comments. We have corrected the sentence of line 54-55 as suggested by the reviewer.
- The introduction needs to be improved with background information on fungal diseases, free radical and problems related to it (with a focus on carcinogenesis) and diabetes as the manuscript focuses on these areas. Also I suggest to start either by describing about "essential oils" or "Zingiberaceae" rather than a generalized description on natural products.
Reply: Thanks for your comments. As per suggestion of the reviewer, we have expanded introduction section in the revised manuscript.
- Why the entire compound from GC-MS not listed out? It is always better to include all the chemicals. In addition, the table should be modified by including the RI/ KI values obtained during experiment as well as based on reference (NIST library).
Reply: Thanks for your comments. We have added all the compounds which show detectable % area in Table of GC-MS. We did not have standards, and alkanes, therefore, we were unable to calculate RI/KI values that time.
- There are several reports on antifungal and antioxidant properties of essential oil from Curcuma longa leaves. What is the significance in repeating the same in the present study?
Reply: Thanks for your comments. Although there are several reports on antifungal and antioxidant properties of essential oil from Curcuma longa leaves, but in our study, we have also incorporated in silico study to validate its claim, and to explain the mechanism of activity of C. longa leaves.
- The authors haven’t mentioned about the statistical tools used in validating the significance of the results.
Reply: Thanks for your comments. We have added statistical tools in results of in vitro antioxidant and anti-diabetic activity as suggested by the reviewer.
- All scientific names should be in italics (Starting from Keywords there were several mistakes)
Reply: Thanks for your comments. We have read the manuscript again and corrected the scientific names in italics as suggested by the reviewer.
- The anti-diabetic activity cannot be taken as a strong indication, especially based on a single assay. I suggest to include the inhibitory potential of alpha glucosidase also.
Reply: Thanks for your comments. We also agree with reviewers comment to include the inhibitory potential of alpha glucosidase, but at this point of time, the activity of essential oil of C. longa of same sample is not possible to perform.
- Though the authors used a common "Results and Discussion", they failed to emphasize the discussion part. Mentioning previous studies don't make the discussion, authors needs to compare the existing results with those previous reports. I suggest to thoroughly modify this section.
Reply: Thanks for your comments. We have modified the discussion part of the revised manuscript as suggested by the reviewers.
Round 2
Reviewer 2 Report
In the paragraph 3.10 authors should include that statistical significance was set at p < 0.05.
Author Response
Thanks for your kind comments. We have added the information as suggested by the reviewer.
Reviewer 3 Report
No more comments to the authors; the article has been improved significantly.
Author Response
Thanks for your kind comments.